Unveiling phenolic content, antibacterial, and antibiofilm potential of sacha inchi (Plukenetia volubilis L.) seed shell extracts against Staphylococcus aureus

Al-Hamoud Gadah A. 1
Amina Musarat mamina@KSU.EDU.SA 1
Alrashoudi Reem Hamoud 2
Mateen Ayesha 2
Maqsood Farah 3
Al-Yousef Hanan M. 1
1 Department of Pharmacognosy, College of Pharmacy, King Saud University , Riyadh , Saudi Arabia
2 Clinical Laboratory Sciences Department, College of Applied Medical Sciences, King Saud University , Riyadh , Saudi Arabia
3 Department of Optometry and Vision Science, College of Applied Medical Sciences, King Saud University , Riyadh , Saudi Arabia
Zothanpuia
Electronic publication date: 2025 Jun 4
Publication date: 2025
Volume: 13
Electronic Location ID: e19524
Received 2024 Nov 18; Accepted 2025 May 5
Copyright: ©2025 Al-Hamoud et al.
Copyright year: 2025
Copyright holder: Al-Hamoud et al.
License: This is an open access article distributed under the terms of the Creative Commons Attribution License, which permits unrestricted use, distribution, reproduction and adaptation in any medium and for any purpose provided that it is properly attributed. For attribution, the original author(s), title, publication source (PeerJ) and either DOI or URL of the article must be cited.
License URL: https://creativecommons.org/licenses/by/4.0/

Keywords: Plukenetia volubilis, Sacha inchi, Phytoconstituent composition, Antioxidant, Antibacterial potency, Antibiofilm activity

Funding: Researchers Supporting Project, King Saud University, Riyadh, Saudi Arabia RSP2025R377 This work was supported by Researchers Supporting Project number (RSP2025R377), King Saud University, Riyadh, Saudi Arabia. The funders had no role in study design, data collection and analysis, decision to publish, or preparation of the manuscript.

==============================
Sacha inchi (SI) seeds are renowned for their high oil content, with omega-3 and omega-6 polyunsaturated fatty acids accounting for approximately 85% of the total fatty acids. However, the use of seed by-products has not received much attention in research. The objective of the current study was to assess the antioxidant and antibacterial properties of aqueous and ethanolic extracts of SI seed shells. The antioxidant potential, along with the total phenolic, flavonoid, and procyanidin content of the aqueous and ethanolic extracts was assessed using the 2,2-diphenyl-1-picrylhydrazyl (DPPH). The ethanolic SI seed shell extract exerted the strongest antioxidant potential, likely due to its higher phenolic and procyanidin content. The antibacterial potency evaluation of extracts towards S. aureus (ATCC29213), S. aureus (clinical) isolate and methicillin-resistant S. aureus (MRSA) demonstrated that the ethanolic extract of SI seed shell possessed significant efficacy. The mean growth inhibition zones of ethanolic extract against tested bacterial strains of S. aureus were ≥ 17.23 ± 0.42 with minimum inhibitory concentration (MIC) values of 250 µg/mL. The time-kill showed the bactericidal effect of ethanolic extract on Gram-positive bacteria, which was demonstrated by the time-kill assay. After ≤ 8 hrs post-inoculation, the mean log reductions in viable bacterial cell counts ranged from 9.37 Log10 to 4.37 log10 CFU/mL for S. aureus (ATCC29213), 9.146 Log10 to 4.124 Log10 CFU/mL for S. aureus (clinical) and 9.367 Log10 to 4.221 Log10 CFU/mL. Also, the ethanol extract exhibited notable potential in reducing biofilm formation and membrane-disruptive properties. Thus, it appears that ethanolic extract of SI seed shells can be potentially used as an excellent source of antioxidants and has antibacterial qualities against certain harmful bacteria that cause infection in the upper respiratory tract and skin.

Introduction

An enormous threat to human society and a serious medical problem worldwide is the genetic ability of pathogenic bacteria to evolve resistance against frequently used antibiotics (Varela et al., 2021). This has made the quest for novel antibacterial compounds from different natural resources, including herbal plants, marine organisms, bacteria, and fungi (Alvarez-Martinez et al., 2021). A wide range of plant-derived secondary metabolites has been widely recognised as effective therapeutic agents in the management of numerous ailments (Chassagne et al., 2021; Alibi, Crespo & Navas, 2021). The ability of some plants to combatpathogenic resistance, has encouraged scientists to isolate and identify active components and investigate their mechanisms (Jubair et al., 2021). Numerous research investigations have been accomplished for the selection of crude plant extracts for the therapeutic cure of pathogenic infections (Ibrahim & Kebede, 2020; Nigussie et al., 2021). The pharmaceutical industries and drug developers are eager to discover and develop structurally novel chemical moieties that are capable of controlling multidrug-resistant pathogenic bacteria. In addition, they are searching for unexplored and underutilized natural resources to produce next-generation drugs. Using plant extracts and phytoconstituents isolated from them can be important therapeutic interventions and could assist in addressing the issue of these drug-resistant microorganisms. Several studies have addressed in the literature that the extracts from Syzygyum joabolanum (jambolan), Punica granatum (pomegranate), Thymus vulgaris (thyme), and Caryophyllus aromaticus (clove) suppressed the growth of the multidrug-resistant Pseudomonas aeruginosa (Pancu et al., 2021; Ahmed et al., 2021; Ahmed et al., 2021; Partovi, Talebi & Sharifzadeh, 2018; Wirtu, Ramaswamy & Maitra, 2024; Nozohour et al., 2018)). The World Health Organisation (WHO) has been an advocate of using medicinal plants as the most reliable source for obtaining a variety of drugs (Dzobo, 2022). According to estimates from the WHO, 80% of people globally already using herbal medicine to treat an array of diseases (Choudhury et al., 2023). This shift towards herbal therapy can be attributed to the accessibility and affordability of herbal plants in comparison to conventional antibiotics. Several plant secondary metabolites including, phenolics, flavonoids, flavones, flavonols, alkaloids, glycosides, and saponins have been used as antimicrobial (Oulahal & Degraeve, 2022), antiviral (Ben-Shabat et al., 2020), antibacterial (Ayele, Akele & Melese, 2022), anti amoebal (Le et al., 2023) and anticanceragainst a variety of microbial strains.

Plukenetia volubilis L. (sacha inchi) is a member of the Euphorbiaceae family, commonly referred as sacha peanut, inca peanut, and mountain peanut. It is native to the Amazon region of Peru and regarded as a sustainable crop with promising economic potential for cultivation in other regions worldwide (Follegatti-Romero et al., 2009). Sacha inchi (SI) bears star-shaped green fruits (3–4 cm), each containing edible, dark blackish-brown seeds (1.5–2 cm) that are slightly swollen at the center and compressed along the edge (Fu et al., 2014). The distinctive SI seed contains a high oil content rich in omega-3, omega-6, and omega-9 fatty acids (35–60%), along with 27% of proteins composed of amino acids like cysteine, threonine, tryptophan, and tyrosine, as well as heat-resistant compounds that contribute to its bitter taste (Fanali et al., 2011; Sathe, Kshirsagar & Sharma, 2012). However, agro-industrialization of SI seeds produces by-products like shells and cake, which are thrown away after initial usage. Various studies suggest that processed by-products derived from various plant sources may serve as significant reservoirs for valuablebioactive components that can offer substantial health benefits (De Camargo et al., 2018). Reusing them would not only produce income for the local community around industrial areas but also lessen the environmental concerns associated with their disposal (Khattak & Rahman, 2017). Scientific interest in agro-wastes generated from industrial food processing has been growing, as they represent a valuable reservoir of bioactive phytoconstituents. Consequently, there is a growing interest in advancing technologies to isolate and harness bioactive extracts from agro-waste for medical applications, due to their potential health-promoting properties (Bala et al., 2023). Various waste materials can serve as raw sources for discovering natural antimicrobials by employing an effective extraction medium and technique. Numerous studies have identified biologically active components with antimicrobial propertiesin various plant-based by-products, including grape seeds and their marc (Krasteva et al., 2023), pomegranate peels (Chen et al., 2020), lemon peels and seeds (Pure, 2023), walnut green husks and shell (Jahanban-Esfahlan & Amarowicz, 2018; Wang et al., 2023), among other sources.

Considering the numerous benefits of vegetables, fruits, and seedwaste by-products, the shells of sacha inchi seeds an industrial by-product may serve as a valuable source of components for medicinal purposes. Previous studies have shown that the SI seed shell is rich in polyphenols and possesses potent antioxidant properties (Chirinos et al., 2016). However, the existing literature on this agro-byproduct lacks detailed information regarding the phytochemical composition of its ethanolic and aqueous extracts, as well astheir biological potencies. Moreover, the relationship between the polyphenolic components present in SI seed shells and their antimicrobial activity remains unexplored.

Therefore, the objectives of the present research were (i) to execute direct analysis of ethanol and aqueous extracts of SI seed shell by HPLC technique (ii) to detect the antioxidant and antimicrobial potential of ethanol and aqueous extracts of SI seed shell and (iii) to anticipate the phytocomponents in charge of antioxidant and antimicrobial activities

Materials and Methods

Plant material

Seeds of sacha inchi were purchased from a local supermarket in Riyadh, Saudi Arabia. Seeds were carefully selected, meticulously washed, and dried. The shell of the seeds was manually separated with the help of a small hammer. Dried shells (500 g) were ground to powder using a professional hammer mill set at 25,000 rpm and 5 min running time at 10 min intervals. The grinding was accomplished at 25 °C for 15 min. The ground fine powder was sifted with a vibratory sieve shaker and sieved to obtain a granulometric fraction of 1.5–2.0 µm diameter. It is well accepted that particles with a finer size produce more extract yield.

Chemicals and standards

The analytical grade solvents and chemicals used for extraction and analysis,including ethanol (99.9%, v/v) phosphoric acid, sodium carbonate (>99%), acetonitrile (100%, HPLC grade), sodium hydroxide and Folin–Ciocalteu reagent (2N solution), were procured from Merk (Darmstadt, Germany). Standards of (+)-catechin hydrate, quercetin hydrate, gallic acid–anhydrous, (-)-epicatechin, p-coumaric, protocatechuic, o-coumaric, caffeic, ferulic, p-hydroxybenzoic acid, chlorogenic, procyanidin (B1, B2, B3 and C1), (-)-epigallocatechin gallate, rutin, resveratrol, and gallic acid glucoside were obtained from Sigma-Aldrich (Darmstadt, Germany). 2,2-diphenyl-1-(2,4,6-tri nitrophenyl)-hydrazinyl (DPPH) and 2,2-azino-bis (3-ethylbenzothiazoline-6-sulphonicacid) diammonium salt (ABTS) used for antioxidant assay, were supplied by Sigma-Aldrich.

Preparation of sacha inchi seed shell extracts

The powdered SI seed shells (five g) were extracted individually with 70% of ethanol (3 × 25 mL) and distilled water (3 × 25 mL) for 3 h at ambient temperature and the boiling temperature (60–80 °C range), respectively. The extraction apparatus consisted of a round-bottom flask connected to a condenser, equipped with a magnetic stirrer and a heating mantle. After 3 h, the resulting organic and aqueous extracts were individually centrifuged for 5 min at 10,000 rpm. The supernatant was separated, the solvent was evaporated under vacuum at ±50 °C temperature using a vacuum evaporator, and extraction yields (%) were determined. The extraction yield percentages were computed by applying the formula, Yield% = W1×100/W2, where W1 andW2 represent the weight of dry extract residue obtained after solvent removal and the weight of the dry seed shell taken. The ethanol and aqueous extract residues were 180.23 and 238.21 mg, respectively. The extract yields were comparable to those of the previously published study (Chirinos et al., 2016). The prepared extracts (ethanol and aqueous) were stored in dark glass vials at 4 °C. Three runs of the extraction process were performed for each extract under similar conditions.

Spectrophotometric determinations

Total phenolic content

The total content of phenols in the SI seed shell extracts (ethanol and aqueous) was estimated by a UV-Vis spectrophotometer (Genesys-5, UV-spectrophotometer) using a colorimetric oxidation/reduction reaction (Singleton & Rossi, 1965). Folin-Ciocalteu reagent was utilized as the oxidizing agent for the analysis. A total of 500 µL of ethanol and aqueous extract diluted extract (20 mg in 10 mL distilled water) were mixed with 2,500 µL of Folin-Ciocalteu reagent (2 N solutions) and 2,000 µL of Na2CO3 solution (75 g/L). The final combination was allowed to sit at an ambient temperature between 20 and 25 °C, for 1 h in the dark. Distilled water was applied as a control. After 1 h standing, absorbance was recorded at 765 nm and TPC was estimated as milligram (mg) of gallic acid equivalent per g of dried test extract (mg GA/g extract). Three replicates were used for each measurement.

Estimation of total flavonoids

An aluminium complexation test was applied to measure the total flavonoid content in the crude SI seed shell extracts (Zhishen, Mengcheng & Jianming, 1999). After diluting one mL test SI seed with 2.8 mL of distilled water and then treated with 0.1 mL each of equimolar 10% aluminium chloride and 1 mg/mL of potassium acetate solutions. The mixture was then allowed to stay in the dark for 30 min and analysed at 415 nm absorbance on a UV-visible spectrophotometer. Distilled water without extract served as the control. The calibration curve was generated by applying quercetin (20–100 µg/mL) as standard. Total flavonoids were analysed from the standard curve, and results were presented as mg of quercetin (QE) equivalents per g of dried shell extract (mg QE/g extract). For every measurement, three replicates were employed.

Estimation of procyanidins

The procyanidin (PC) concentration in the prepared SI seed extracts was analysed by obeying the modified Sun, Ricardo-da Silva & Spranger (1998) method. A 400 µg/mL concentration of extract was prepared in the methanol. Absorption (A) was used to develop a calibration for catechin solutions with 30 to 600 µg/mL concentration range. Varied combinations were prepared, incubated for 30 min at 30 °C, and then their absorbance was noted at 500 nm on a spectrophotometer. Methanol without extract was used as a control. The calibration curve was employed to compute the total procyanidin concentration in each investigated sample and results were represented as mg of catechin (CAT) equivalents per g of dried SI seed shell extract (mg CAT/g extract). Three replicates were used for each measurement.

Estimation of anthocyanins

The total anthocyanin content present in the test extracts was quantified following the method described by Lee et al. (2005), with minor modifications. 300 µL of test extract was diluted with buffer solution (pH 1) to achieve 1.1 absorbance and the corresponding dilution factor was measured. The extract was then diluted using the same dilution factor with a buffer solution (pH 4.5). Two absorbance at 520 and 700 nm were recorded using a spectrophotometer. The anthocyanins (AC) content was expressed as mg equivalent of cyaniding-3-glucoside per g of dried shell extract (mg AC/g extract) and computed by applying equation (AC = A.—M.F. 103/ɛ. 1, mg/g). Where A, M, F, 103, 1, and ɛ represents absorbance, molecular formula of cyaniding-3-glucoside, dilution factor, factor for converting from g to mg, path length in cm and 26,9000 molar extinctions co-efficient in L/mol. cm. Each measurement was performed in three repetitions, with extract-free buffer solution serving as the control.

HPLC analysis

The HPLC analysis of ethanolic and aqueous extracts of SI seed shellswas conducted on an LC-20AD Shimadzu HPLC system (Shimadzu, Kyoto, Japan), equipped with a binary solvent delivery module (LC-20AD), a Rheodyne type injector with a 20 µL sample loop and a DAD detector (SPD-M 20 A). An extended guard column was used to perform reverse phase column chromatographic separation (C-18 Pack Capcell column with five µm, 250 mm × 4.6 mm dimensions). The eluent system constitutes methanol-acetonitrile water (MeOH: ACN: H2O: 40:15:45 v/v) with 1% acetic acid, using isocratic elution over a 30 min period. The DAD was set to operate within a wavelength range of 240 to 800 nm. 20 µL of samples and reference solutions were applied, with flow rate maintained at one mL/min. Software Shimadzu LC solution was employed to collect and analyse the data. The phytocomponents present in the extracts were identified on the basis of UV absorbance (between 240 and 800 nm), retention times for the analytes, and reference standards.

Antioxidant activity

The antioxidant properties of ethanolic and aqueous extract of SI seed shell was elucidated by DPPH assay as described by Katalinic et al. (2006). The antioxidant potentials observed in DPPH radical-scavenging were attributed to the compound’s ability to donate hydrogen atoms. The DPPH• free radical accepts readily an electron or hydrogen atom to become a stable molecule. The antioxidant activity was assessed by monitoring the reduction in absorbance of the DPPH• radical at 517 nm, which was accompanied by a colour transition from violet to yellow. The ability of SI seed shell extracts to scavenge stable DPPH• radicals was assessed using spectrophotometric analysis. The solutions of extract were prepared by dissolving 20 mg of dry extract in 40 mL of methanol. Prior to UV measurements, a DPPH• solution was prepared in methanol (6 × 105 M). After that, three mL of DPPH• solutions were mixed with varied doses (62.5, 125, 250, 250, 500, 1,000 µg/mL) of each test extract and placed under dark conditions at ambient temperature (25 °C) for 30 min. After 30 min, absorbance at 517 nm was recorded. Ascorbic acid was applied as standardand methanol as control. Triple execution and observation of the experiments were carried out. The radical scavenging percentage was computed by applying following equation. Scavenging%=AC−ASAC×100

where Ac and As denoted absorbance of the control and test sample, respectively.

Antibacterial activity

Bacterial strains

Three strains of Staphylococcus aureus (S. aureus) were utilized in this study: S. aureus (ATCC29213), S. aureus (clinical) and methicillin-resistant S. aureus (MRSA). MRSA and S. aureus clinical isolates were procured from the King Fahad Medical City, Riyadh, Saudi Arabia and S. aureus (ATCC29213) was available in our laboratory. S. aureus is a genetically diverse pathogen, with different strains displaying a variety of phenotypic traits such as antibiotic resistance, virulence factors, and pathogenic mechanisms. These strains show a broad spectrum of resistance, including MRSA. Evaluating multiple strains of S. aureus is essential to ensure that the outcomes of antimicrobial interventions or therapeutic strategies are applicable to both resistant and non-resistant strains, which is vital for the development of effective treatments.

Evaluation of antibacterial activity

The antibacterial effect of ethanolic and aqueous extracts of SI seed shell was performed using an agar well diffusion assay (Hassan & Ullah, 2019). Briefly, the agar plate surface was inoculated by three strains of S. aureus at a concentration of 1.5 × 108 CFU/ml, over the entire agar surface. Then, eight mm of the circular hole was made aseptically into the agar plate using a sterile cork borer, and a volume of 100 µL of the test sample stock (40 mg/mL) solution was introduced to the well. The positive control employed was the standard antibiotic Gentamycin (10 µg/mL), while water served the negative control. The inhibition zone was subsequently assessed by incubating all of the prepared agar plates for 24 h at 37 °C.

Determination of minimum inhibitory concentration

The minimum inhibitory concentration (MIC) of test samples was ascertained by broth microdilution assay by preparing stock solution (40 mg/mL), conducted in 96-well polystyrene sterile plates towards S. aureus (ATCC29213), S. aureus (clinical) isolated, and MRSA strains. The preparation of microdilution plate included putting in 100 µL of Mueller Hinton Broth (MHB) to wells 2 through 12. The first well contained 200 µL of test sample only and a serial two-fold dilution was performed from wells 2 to 10 by transferring 100 µL from the preceding well to the next, making the total final volume of 100 µL in each of the wells. Lastly, all wells except wells 1 and 12 received 10 µL of bacterial suspension. A negative control with only MHB was designated for well 12, whereas well 11 were retained as a positive control with MHB plus bacterial suspension but no test sample. After that, plates were incubated for 18 to 24 h at 37 °C. After incubation, 40 µL of a freshly prepared 3-(4,5-dimethylthiazol-2-yl)-2,5-diphenyltetrazolium bromide (MTT) reagent (0.5 mg/mL in sterile water) was added to each well (2 to 12), resulting in a total volume of 150 µL in wells 2 to 11, and 140 µL in well 12. The resultant mixture was further incubated (37 °C, 30 min). Following incubation, bacterial growth inhibition was assessed by observing the reduction of yellow MTT to purple formazan crystals by metabolically active cells. The MIC of extract was defined as the minimal extract concentration that inhibited apparent bacterial growth, including that of the standard strains. Gentamycin was employed as positive control and each assay was conducted three times (Xu et al., 2023).

Time kill kinetics

The time-kill kinetics and dose-dependent kinetics test of SI seed shell extracts towards S. aureus (ATCC29213), MRSA, and S. aureus (clinical) strains have been established using a modified time-kill kinetics approach (Appiah, Boakye & Agyare, 2017). Different concentrations (62.5 (1/4MIC), 125 (1/2MIC), 250 (1MIC), 500 (2MIC), and 1,000 (4MIC)) µg/mL) of shell extracts were used against test bacteria by obeying the same protocol as used for evaluating MIC. Microbial growth was estimated by plating 10-fold serial dilutions on MHA after 0, 2, 4, and 8 h of incubation at 37 °C. Three duplicates of each test were conducted. Log10 CFU/mL was applied to express the results.

Antibiofilm assay

The antibiofilm assay was performed by using varying concentrations (250, 500, and 1,000 µg/mL) of ethanolic SI seed shells extract. TSB broth was utilized to culture S. aureus (ATCC29213), MRSA, and S. aureus (clinical) strains for the entire day at 37 °C. To perform microtitration, 96-well polystyrene plates were filled with 200 µL of bacterial suspensions in triplicate, followed by the addition of various doses of SI seed shell extracts (62.5, 125, 250, 500, and 1,000 µg/mL), except for one well which served as an untreated control (UT). Plates were then incubated for a period of 24 h at 37 °C. Suspensions were taken out after incubation, and wells were given three PBS washes (pH 4.0). The wells were treated with 200 µL of glacial acetic acid (30%) for 15 min at ambient temperature after being stained for 30 min with 200 µL of crystal violet (0.1%) solution and then rinsed with PBS. Afterwards, 100 µL of the dispersed crystal violet has been moved to a flat-bottom microtiter dish and absorbance at 595 nm was recorded on ELISA reader. Biofilm inhibition levels were calculated relative to the biofilm formed in untreated wells (set as 100%) and sterile media controls (set as 0%). The results were averaged across three independent biological replicates (Alrashoudi et al., 2025)

Determination of antibiofilm activity by inverted microscope

The antibiofilm activity was detected using the microscopy technique. The three S. aureus strain cultures received individual treatments with varying doses (250, 500, and 1,000 µg/mL) of ethanolic extract of SI seed using a 12-well microtiter plate. One well of a 12-well microtiter plate was left untreated as a control, while Gentamycin treatment served as the positive control. The biofilms were examined using an inverted microscope in a 12-well microtiter plate at 40 × magnification following a 24 h incubation period at 37 °C (Alangari et al., 2022).

Assessment of the cell membrane integrity

Three strains of S. aureus were assessed for cell membrane integrity based on the amount of cell material that leaked into the medium, including proteins and nucleic acids (Nguyen et al., 2024). The cultures of S. aureus were incubated with varied doses of SI seed shell extracts (62.5, 125, 250, 500, and 1,000 µg/mL) for 24 hat 37 °C. The supernatant was separated from the suspensions by centrifuging them for 5 min at 13,000 rpm. The level of liberated proteins and nucleic acids were analysed by using a nano-drop at absorbance of 260 nm and 280 nm on a spectrophotometer, respectively. An untreated sample was retained as a negative control, and Polymyxin B served as the positive control.

Statistical analysis

GraphPad Prism 8.4.3 was applied to execute statistical analyses. Brown-Forsythe ANOVA was utilized to evaluate the significant differences in dose-dependent biofilm inhibition and membrane disruption by SI seed shell extracts across various S. aureus strains. Regression equations derived by plotting the graph between inhibition percentage (%) and concentration were used to determine the IC50 value. The significance threshold of p < 0.05 was established.

Ethical approval

This study was authorized by King Fahad Medical City’s Institutional Review Board in Saudi Arabia and performed in accordance with the Declaration of Helsinki’s standards (IRB log number: 22-026E).

RESULTS AND DISCUSSION

Chemical composition and antioxidant potential of SI seed shell extract

The extraction of bioactive components from the plant material largely depends upon the solvent type used during the extraction process. Ethanol, methanol and water are the solvents most commonly employed in studies investigating the antibacterial properties of plants (Truong et al., 2019). The high extract yields can be achieved only by selecting most suitable extraction solvent/solvent mixture based on the characteristics of herbal material. Two extraction solvents ethanol (70%) and water were used for the extraction of the SI seed shell and the yield of each extract was determined. The results showed that 70% ethanol produced the highest extraction yield at 17.43 ± 0.063%, significantly higher than the 10.27 ± 0.014% yield obtained with water extraction.

To examine the correlation between the polyphenolic attributes of the extract and their antioxidant and antibacterial efficacies, the total polyphenolic content (TPC), total flavonoids (TF), procyanidins (PC), and anthocyanins (AC) content in the prepared extracts were measured by spectrophotometric technique (Table 1). The results exhibited that the 70% ethanolic extract of the SI seed shell had a significantly higher content of total phenolic content (113.21 ± 1.04) and total flavonoid content (43.17 ± 0.94) in contrast to the aqueous extract of the SI seed shell. Similarly, the estimation of procyanidin content revealed that the ethanolic extract of the SI seed shell contained higher amounts of procyanidins (174.12 ± 1.01) than the aqueous extract (152.36 ± 0.39). However, lesser amounts of anthocyanins were found in both extracts. The extraction of phenols, flavonoids, and procyanidins was found to be more significant with ethanol than with water. This might be because the ethanol has better solubility for these phytoconstituents. These findings corroborate previous research demonstrating that alcoholic solvents like ethanol and methanol are more efficient than water at extracting the chemical components from medicinal plants (Liaudanskas et al., 2021; Stanciauskaite et al., 2021).

Table 1 Spectrophotometric assessment of phenolic content of ethanol and aqueous extract of SI seed shell.

SI seed shell	TPC (mg GA/g extract)	TF (mg QE/g extract)	PC (mg CAT/g extract)	AC (mg/g extract)	
Ethanol extract	113.21 ± 1.04	43.17 ± 0.94	174.12 ± 1.01	9.15 ± 1.02	
Aqueous extract	64.57 ± 1.72	32.51 ± 0.57	152.36 ± 0.39	3.46 ± 1.31	
Notes.

± The triplicate experimental results.

The individual constituents in the ethanolic and aqueous extract of SI seed shells were identified and quantified by HPLC. The HPLC chromatograph of chemical components and extracts under investigation were depicted in Figs. 1A and 1B. Standard calibration was the technique that was used for quantification. The following order of components was obtained upon elution under the specified HPLC conditions. The amount of individual components in the SI seed shell extracts, as determined by HPLC was presented in Table 2. In the ethanolic extract of SI seed shells, cinnamic acid derivative (12.14 ± 2.36 mg/g) was the most abundant component, followed by hydroxycinnamic acid derivative (10.26 ± 1.03 mg/g), cinnamic acid (7.21 ± 0.92 mg/g), procyanidin B1(6.34 ± 0.94 mg/g), procyanidin B2 (5.69 ± 0.51mg/g) and protocatechuic acid (4.12 ± 1.32 mg/g). In contrast, aqueous extract of SI seed shells contained a high content of procyanidin B1 (8.78 ± 0.8736 mg/g), along with appreciable amounts of hydroxyl cinnamic acid derivative, cinnamic acid derivative, and (+)-catechin. As estimated by HPLC, the ethanolic extract of SI seed shells had the highest TPC compared to aqueous extract. It is evident that there is a strong association between the results of the spectrophotometric and chromatographic techniques. All the quantified phenols in the extracts have therapeutic applications.

Figure 1 HPLC chromatogram of (A) ethanol and (B) aqueous extract of SI seed shell.

Table 2 HPLC determination of phenolic compounds in ethanol and aqueous extract of SI seed shell.

Peak	Ethanol extract	Aqueous extract	
	Retention
time (min)	Compound	Amount	Retention
time (min)	Compound	Amount
(mg/Kg)	
1	3.32	Quinic acid	0.24 ± 0.01	2.91	Quinic acid	0.12 ± 0.02	
2	5.23	Gallic acid	0.82 ± 0.04	5.11	Gallic acid	0.20 ± 0.03	
3	9.87	Cinnamic acid	7.21 ± 0.92	9.11	Cinnamic acid	2.14 ± 0.27	
4	12.9	Cinnamic acid
derivative	12.14 ± 2.36	13.16	Cinnamic acid
derivative	5.27 ± 0.29	
5	15.1	p-Coumaric acid	2.89 ± 1.04	15.92	p-Coumaric acid	ND	
6	17.02	Ferulic acid	0.67 ± 0.04	18.16	Ferulic acid	ND	
7	19.90	Protocatechuic acid	4.12 ± 1.32	20.17	Protocatechuic acid	3.25 ± 0.11	
8	21.32	(+)-Catechin	1.48 ± 0.64	22.13	(+)-Catechin	4.37 ± 1.75	
9	23.2	Hydroxycinnamic
acid derivative	10.26 ± 1.03	24.15	Hydroxycinnamic
acid derivative	5.82 ± 0.96	
10	42.73	Procyanidin B3	2.51 ± 0.67	42.11	Procyanidin B3	3.46 ± 0.28	
11	43.81	Procyanidin B1	6.34 ± 0.94	44.11	Procyanidin B1	8.78 ± 0.87	
12	49.87	m-Coumaric acid	1.23 ± 0.11	46.91	m-Coumaric acid	ND	
13	50.92	Procyanidin B2	5.69 ± 0.51	50.23	Procyanidin B2	3.41 ± 1.34	
14	51.73	Procyanidin C1	0.26 ± 0.16	51.92	Procyanidin C1	0.78 ± 0.51	

Antioxidant potential

The antioxidant capacity of SI seed shell extracts was measured using the DPPH radical scavenging technique. The results of the antioxidant activity of ethanol and aqueous extract of SI seed shells are shown in Table 3. The ethanolic extract of SI seed shells demonstrated significantly higher DPPH radical scavenging capabilities at all tested concentrations compared to the aqueous extract of SI seed shells (P < 0.05, Table 3). The antioxidant potential of SI seed shell extracts was found to be dose-dependent. However, compared to the DPPH• radical scavenging activities of SI seed shell extracts, the standard (ascorbic acid) showed noticeably greater actions. This study also computed the IC5 0 values, representing the doses of the investigated plant extracts required to scavenge 50% of the DPPH radicals. The IC50 values for ethanol and aqueous extract were 31.05 and 66.24 µg/mL, respectively. Conversely, the standard (ascorbic acid) has an IC50 value of 13.32 µg/mL. The higher antioxidant capacity of SI seed shells may be may be attributed to the substantial presence of phenolic and flavonoid components in the ethanolic extract. The variation in the antioxidant capacity of ethanol and aqueous extract may be due to significant variances in their phenolic contents. Also, the extraction solvent has an impact on the extract’s ability to scavenge free radicals. The findings suggest that ethanol would be the best solvent choice for extracting antioxidant components. Overall, the ethanolic extract from SI seed shells showed the highest polyphenol content and antioxidant capacity. The antioxidant action of phenolic components, which allow them to act as hydrogen donors, reducing agents, and singlet oxygen quenchers, are primarily attributed to their redox characteristics. They might also have the ability to chelate metals (Hossain et al., 2011). The main factor influencing the biological action of polyphenols is the presence of at least one phenolic hydroxyl (OH) group in their structure. Thus, key factors affecting antioxidant capacity include the number and position of hydroxyl groups on the aromatic ring, as well as presence of methoxy substituents in the ortho position relative to the hydroxyl group (OH) (Anouar et al., 2013; Rice-Evans, Miller & Paganga, 1996). Additionally, flavonoids also have an antioxidant effect, but this effect varies from among the molecules and is determined by the degree of hydroxylation and the placement of the -OH groups in the B ring. In particular, a B ring with an ortho-hydroxylated structure exhibits enhanced activity because it acts as a preferential binding site for trace metals (Pietta, 2000) or gives the aroxyl radical more stability through the delocalization of electrons (Van Acker et al., 1996).

Table 3 In vitro DPPH scavenging activities of ethanolic and aqueous extracts of SI seed shell.

Concentration (µg/mL)	DPPH scavenging activity (% inhibition)	
	Ascorbic acid	Ethanol extract	Aqueous extract	
62.5	35.70 ± 1.14*	28.27 ± 1.11**	24.17 ± 1.01**	
125	51.17 ± 0.96*	42.26 ± 0.86**	27.12 ± 1.01**	
250	62.95 ± 0.98*	55.56 ± 0.72*	43.54 ± 0.76**	
500	70.24 ± 0.21*	69.36 ± 0.89*	52.41 ± 0.98**	
1,000	76.13 ± 1.16*	71.62*±1.7*	57.25*±0.91**	
IC50	13.32	31.05	66.24	
Notes.

The values indicated are mean ± SD (n = 3); Significant differences between the groups are indicated by values with an asterisk (*) at P < 0.05. At p < 0.05, the values with ** are significantly different.

Figure 2 (A–C) Well diffusion assay showing antibacterial activity of 1-ethanol SI seed extract and 2-aqueous SI seed extract against different strains of S. aureus.

Antibacterial activity

The characterized extracts (ethanol and aqueous) of SI seed shells were tested for antibacterial against the three strains of S. aureus including S. aureus (ATCC29213), S. aureus clinical, and MRSA. Agar diffusion and a test to ascertain the MIC were the two methods employed for this purpose. The impact of SI seed shell extracts on the growth of S. aureus (ATCC29213), S. aureus (clinical) and MRSA was evaluated. Fig. 2, presents the results of tested bacterial strains treated with ethanol and aqueous extracts of SI seed shells. The antibiotic Gentamycin served as a positive control was examined on a separate Petri dish. The results of the antibacterial effect revealed that the ethanolic extract of SI seed shells exerted strongsensitivity towards S. aureus (clinical) strain with a high (20 mm) zone of inhibition (Fig. 2A), while the significant influence on S. aureus (ATCC29213) (17 mm) and MRSA (17 mm) with a similar zone of inhibition (Figs. 2A and 2B), at 40 mg/mL concentration (Table 4). However, aqueous extract of SI seed shells demonstrated lower inhibition zones and moderate antibacterial activity against all tested strains of S. aureus at the same dose. The antibiotic Gentamicin, at 10 µg/mL dose, demonstrated antibacterial effect comparable to that of the aqueous extract of SI seed shells. The results showed that the ethanolic SI seed shells extract exhibited greater antibacterial potential than both the aqueous extract and the positive control (Gentamicin), effectively targeting all three S. aureus strains studied. The discrepancies in the antibacterial properties of the ethanol and aqueous extracts of SI seed shells observed in this study may be attributed to variances in their phytochemical composition. The enhanced antibacterial action of the ethanolic SI seed shells extract may be due to the substantial presence of phytoconstituents such as phenols, flavonoids, terpenoids, and tannins. These results are consistent with previous studies that reported the antimicrobial potential of plant extracts rich in bioactive compounds, which target a broad spectrum of bacterial strains by disrupting essential bacterial functions. The phenols are reported to interfere with the cytoplasmic membrane function, disrupting energy metabolism and impacting nucleic acid synthesis (Salehi-Sardoei & Khalili, 2022). It has been demonstrated that flavonoids block the bacterial enzymes RNA polymerase, reverse transcriptase, telomerase, and DNA polymerase (Kouadri, 2018). Also, plant extracts have a greater effect on Gram-positive bacteria (S. aureus) than Gram-negative bacteria. Structural differences in their cell walls may make Gram-positive bacteria more vulnerable.Gram-negative bacterial cells possess an additional outer membrane that provides a hydrophilic surface, serving as a barrier to the permeability of many substances, including biological molecules (Tavares et al., 2020).

Table 4 Determination of diameter of zone of inhibition of ethanol and aqueous extract of SI seed shell.

S. No	Bacterial strains	Zone of Inhibition (mm)
Ethanol extract	Zone of Inhibition(mm)
Aqueous extract	MIC
(µg/mL)	
1	S. aureus (ATCC29213)	17	14	250	
2	S. aureus (Clinical)	20	14	250	
3	MRSA	17	15	250	

Minimum inhibitory concentration

The lowest dose of an antimicrobial drug that can prevent the discernible rise in microbial growth following an overnight incubation is known as the minimum inhibitory concentration (MIC). The MIC was determined to evaluate the antibacterial efficacy of the prepared SI seed shell extract at reduced antimicrobial concentrations. Antimicrobials with low MIC values are typically more potent, while those with high MIC values are usually less effective. The MIC value of ethanol and aqueous extract of SI seed shellswas studied and was found to be 250 µg/mL for all the three investigated strains of S. aureus (Table 4). The MIC for all strains (250 µg/mL) further reinforces the efficacy of SI seed extracts as an antibacterial agent, consistent with findings from other studies on plant extracts exhibiting similar MIC values towards Gram-positive bacteria (Hemeg et al., 2020). Thus, the ethanolic extract of SI seed shell with higher antibacterial activity performance and lowest MBC/MIC ratio was examined for the time-kill behaviour and antibiofilm activity.

Time-kill kinetics

The bactericidal activity of ethanolic extract of SI seed shell was estimated by assessing the time course to kill the tested bacteria. The time-kill kinetics of SI seed shell ethanol extract was conducted at five varied doses (62.5–1,000 µg/mL) towards three bacterial strains of S. aureus (ATCC29213), S. aureus (clinical) and MRSA strains over a period of 8 h. Consequently, time-kill graph was plotted between the logarithmic numbers of CFU/mL versus incubation period. The number of surviving bacteria in the extract was determined at the time of sampling using the plate count technique. The percentage decrease and logarithmic reduction in the microbial population at each time point were calculated to represent changes (either reduction or growth) relative to the initial inoculum. The population of test organisms decreased significantly (p ≤ 0.05) at each interval when tested at doses of 1,000 µg/mL and 62.5 µg/mL in the time-kill kinetics investigation against S. aureus, as indicated in Fig. 3. The time kill kinetics assay of SI seed shell ethanol extract showed time and dose-dependent activity against S. aureus strains. The inhibition of S. aureus (ATCC) strain with varying doses (62.5–1,000 µg/mL) of ethanolic SI seed shell extract at successive time intervals (0 h, 2 h, 4 h, and 8 h) showed reduction in Log10 CFU/mL from 9.37 Log10 CFU/mL to 4.37 log10 CFU/mL, indicated dose and time dependent bactericidal activity (Fig. 3A). However, S. aureus clinical and MRSA strainsas demonstrated in Figs. 3B and 3C, displayed a notable decline (p ≤ 0.05) in the test organism population was equally noticed at each time interval when tested atdifferent concentration. In comparison to untreated (UT) S. aureus strains, the S. aureus (clinical) showed an average log reduction in viable cells between 9.146 Log10 and 4.124 Log10 CFU/mL, while, the MRSA strain displayed a 9.367 Log10 to 4.221 Log10 CFU/mL reduction in viable cells. Thus, time-kill kinetics test verified that both time and dose affected thebactericidal activity of SI seed shell ethanol extract. This demonstrates that the SI seed shell extract can both supress bacterial growth and effectively kill the bacteria, which is crucial for therapeutic applications. These findings supported earlier studies that demonstrated the bactericidal effects of plant-derived antimicrobials in a time and dose-dependent manner, indicating that prolonged exposure to bioactive plant components can effectively lower bacterial populations (Techaoei, 2022). Numerous bioactive substances with antibacterial qualities, such as polyphenols, flavonoids, and fatty acids, are present in the ethanolic extract of SI seed shell. The bioactive components are capable of disrupting bacterial cell membranes, allowing internal contents to leak out and limiting cell viability. The flavonoids and polyphenols in the extracts tend to bind to bacterial proteins, leading to their denaturation or inactivation, thereby impairing bacterial survival and function. Additionally, these components may inhibit transpeptidase and glycosyltransferases, two key enzymes involved in peptidoglycan synthesis, thereby weakening the cell wall and leading to cell lysis (Fig. 4) (Vaou et al., 2021).

Figure 3 Time kill kinetics effect of ethanol extract of SI seed shell on (A) S. aureus (ATCC), (B) S. aureus (clinical), and (C) MRSA at different doses and time.

Time kill kinetics effect of ethanol extract of SI seed shell on (A) S. aureus (ATCC), (B) S. aureus (clinical), and (C) MRSA at different doses and time.

Figure 4 Effect of SI seed shell extract against S. aureus antibacterial mechanism inhibition of quorum sensing by blocking the binding of auto-inducing peptides (AIPs) to the AgrC receptor interferes with biofilm formation by inhibit.

Figure 5 Antibiofilm activity of ethanol extract of SI seed shell on (A) S. aureus (ATCC), (B) S. aureus (clinical), and (C) MRSA.

Significant reduction in biofilm formation with increased concentrations of the extract, with the greatest red. **** represents the significance (p ≤ 0.0001).

Antibiofilm activity

Biofilm formation, a significant factor in the persistence of bacterial infections and antibiotic resistance, was effectively inhibited by SI seed shell ethanol extract. The ethanol extract of SI seed shell exhibited significant antibiofilm activity, with biofilm formation decreasing progressively as the extract concentration increased. The antibiofilm activity of treated S. aureus (ATCC29213) strain exerted a significant decline in the antibiofilm activity as the concentration of the extract increased (p ≤ 0.0001) and showed a decline in OD values from 0.878 to 0.451, in contrast to untreated S. aureus (ATCC29213) strain with 1.228 OD value (Fig. 5A). The ethanol extract of SI seed shell efficiently inhibits the biofilm of the other two strains S. aureus (clinical) and MRSA. The extract displayed similar dose-dependent antibiofilm activity with a decrease in OD values from 0.769 to 0.379 for the biofilm of S. aureus clinical (Fig. 5B), whereas extract-treated MRSA biofilm showed a decrease in OD values from 0.671 to 0.319 (Fig. 5C), as compared to untreated S. aureus clinical (OD: 0.781) and untreated MRSA (OD: 0.766). The dose-dependent decrease in biofilm formation, as evidenced by OD reductions for S. aureus (ATCC29213), S. aureus (clinical), and MRSA strains, supported the hypothesis that the extract interferes with biofilm development. This is a significant finding, as biofilms are notoriously resistant to conventional antibiotics treatments. The influence of varied ethanol extract concentrations of SI seed shell on the three tested strains of S. aureus biofilm formation was further validated by the inverted microscopy images. All three of the untreated S. aureus strains showed extensive biofilm formation in their SEM images (Fig. 6A). However, the SEM monographs of extract-treated S. aureus (ATCC29213), S. aureus (clinical), and MRSA strains showed a progressive decrease in biofilm density as the extract concentration increased (Fig. 6B). The most significant decrease in biofilm formation was noted at the highest dose (1,000 µg/mL). Previous studies have provided strong evidence that medicinal plants have the potential to treat a range of viral diseasescaused by numerous pathogens (Manandhar, Luitel & Dahal, 2019). The antibacterial potential of medicinal plants make them effective in curing a variety of diseases (Karygianni et al., 2016). Numerous phytoconstituents found in medicinal plants have been demonstrated to inhibit of quorum sensing molecules and biofilm formation in addition to playing a significant role in preventing microbial pathogenicity (Uc-Cachon et al., 2021). Plant extracts and isolated components have been shown to stop or interfere with the production of biofilms by blocking quorum sensing pathways or directly harming the biofilm matrix; these mechanisms may also be important in the current work with SI seed shell ethanol extract (Uc-Cachon et al., 2021). The ethanolic SI seed shell extract appears to interfere with several of the critical molecular mechanisms that S. aureus uses to build biofilms. One key target is the Agr system, which controls quorum sensing and biofilm development. The Agr system is activated by auto-inducing peptides (AIPs) that bind to the AgrC receptor. This AIP bindingmay be inhibited by the bioactive substances in SI seed shell ethanol extract, which would stop the subsequent processes that promote the production of biofilms. The inhibition of agrA and agrC could result in the suppression of virulence factor expression (Gray, Hall & Gresham, 2013; Cosgriff et al., 2019). Additionally, the synthesis of polysaccharide intercellular adhesin (PIA) is essential for biofilm formation in S. aureus. The ica operon, comprising the genes icaA, icaD, icaB, and icaC, is responsible for producing PIA and other biofilm matrix components, with icaA and icaD being directly involved in PIA biosynthesis (Tamai et al., 2023). The ethanolic extract of SI seed shell may interfere with the expression of these ica genes, particularly icaA and icaD, leading to a reduction in PIA production. By decreasing the expression of these genes and destabilizing the biofilm matrix, the extract may effectively inhibit biofilm formation (Fig. 4) and thereby enhance the effectiveness of antimicrobial agents (Lu et al., 2019). Moreover, the extract may increase S. aureus sensitivity to oxidative stress, potentially by inducing the production ofreactive oxygen species (ROS) that compromisebiofilm integrity and reduce bacterial viability (Guo et al., 2023).

Figure 6 Inverted microscopy images of (A) biofilm of S. aureus (ATCC), S. aureus (clinical) and MRSA, and (B) showing a visible reduction in biofilm density as the SI seed shell extract concentration increased in all the tested strains.

Membrane disruption and nucleic acid leakage

The results of cell membrane integrity measurements showed a gradual increase of protein release over time in the presence of SI seed shell ethanol extract. As the extract concentration increased, the treated S. aureus (clinical) strain exhibited a significantly higher level of protein leakage from the bacterial cells (from 0.022 to 0.058 µg/mL); in contrast, untreated S. aureus clinical strain showed only 0.015 µg/mL of protein in the medium (Fig. 7B). A similar increase in protein leakage level from the bacterial cells of treated strains of S. aureus (ATCC29213) strain (rising from 0.018 to 0.055 µg/mL) and MRSA (rising from 0.015 to 0.054 µg/mL)in the culture medium was observed with increasing concentration of SI seed shell ethanol extract (Figs. 7A and 7C). Additionally, a comparable concentration-dependent effect was noted in thenucleic acid leakage from the bacterial cells of S. aureus (ATCC29213) strain exposed to the SI seed shell ethanol extract. The concentration of SI seed shell extract was found to increase nucleic acid leakage in the treated S. aureus strains, indicating its effectiveness in disrupting bacterial membranes. The highest level of nucleic acid leakage (1.867 mg/mL) in the treated S. aureus (ATCC29213) strain was observed at 1,000 µg/mL concentration of SI seed shell ethanol extract (Fig. 8A), indicating a significant rise in membrane permeability with rising extract doses (p ≤ 0.0001). In comparison, at similar concentrations, the S. aureus (clinical) strain and the antibiotic-resistant strain MRSA exhibited notable nucleic acid leakage of 1.794 mg/mL and 1.523 mg/mL, respectively (Figs. 8B, and 8C), demonstrating the consistent efficacy of SI seed extract against both sensitive and resistant strains. Thus, all the three strains of S. aureus treated with the ethanol extract of SI seed shellshowed significant leakage ofproteins and nucleic acids, suggesting that the extract effectively disrupts bacterial membranes, a key mechanism behind its antibacterial action. This membrane-disruptive effect of SI seed shell extract aligns with the findings of other studies on plant-based antimicrobials that have shown similar capabilities to permeabilize bacterial membranes, leading to cell lysis and death (Lin et al., 2021). Membrane disruption is a critical antibacterial strategy, as it causes the loss of cellular contents and death, which is essential for combating antibiotic-resistant strains like MRSA. These findings underscore the importance of developing therapies that target bacterial membranes, especially in the context of rising antibiotic resistance.

Figure 7 Protein leakage induced by SI seed shell extract in (A) S. aureus ATCC, (B) S. aureus clinical and (C) MRSA.

A dose-dependent protein leakage was observed, with a maximum at 1,000 µg/mL extract concentration, indicated membrane disrupt. **** represents the significance (p ≤ 0.0001).

Figure 8 Nucleic acid leakage induced by SI seed shell extract in (A) S. aureus ATCC, (B) S. aureus clinical and (C) MRSA.

A dose-dependent protein leakage was observed, with a maximum at 1,000 µg/mL extract concentration, indicated membrane disrupt. **** represents the significance (p ≤ 0.0001).

CONCLUSION

This study contributes to the research underway in response to the growing desire to develop alternatives based on natural substances with antibacterial qualities, such as extracts from SI seed shells. The aqueous and ethanol extracts of the SI seed shell were prepared and evaluated antioxidant and antibacterial potential. The highest content of polyphenols and antioxidant capacities were recorded for the ethanol extract of the SI seed shell. The antibacterial activity evaluation of extracts against three strains of S. aureus (S. aureus (ATCC29213), S. aureus (clinical), and MRSA), ethanolic extract of SI seed shell has been proven to be highly effective. The promising antioxidant and antibacterial potential exerted by the ethanol extract of SI seed shell suggested that it has significant potential as an alternative or adjunctive treatment for S. aureus infections, including those caused by antibiotic-resistant strains. Its time-dependent bactericidal action, inhibition of biofilm formation, and capacity to rupture bacterial membranes made it an effective therapeutic agent for the future. These findings necessitate in vivo antibacterial research of this natural antimicrobial agent.Future research should focus on standardization of extraction techniques, evaluation of plant extracts against a broader range of clinically relevant drug-resistant pathogenic isolates, isolation, identification, and characterization of active constituents to locate the antibacterial potential as well as in vivo studies investigations, or clinical trials of the plant extracts/isolated components to assess the extracts’ potential for therapeutic use.

Supplemental Information

Supplemental Information 1 Spectrophotometric assessment of phenolic content of ethanolic and aqueous extract of SI seed shell

Supplemental Information 2 HPLC determination of phenolic compounds in ethanol and aqueous extract of SI seed shell

Supplemental Information 3 In vitro DPPH scavenging activities of ethanolic and aqueous extracts of SI seed shell

Additional Information and Declarations

Competing Interests

Author Contributions

Data Availability

The authors declare there are no competing interests.

Gadah A. Al-Hamoud analyzed the data, authored or reviewed drafts of the article, and approved the final draft.

Musarat Amina conceived and designed the experiments, authored or reviewed drafts of the article, and approved the final draft.

Reem Hamoud Alrashoudi analyzed the data, authored or reviewed drafts of the article, and approved the final draft.

Ayesha Mateen performed the experiments, authored or reviewed drafts of the article, and approved the final draft.

Farah Maqsood analyzed the data, prepared figures and/or tables, and approved the final draft.

Hanan M. Al-Yousef analyzed the data, authored or reviewed drafts of the article, and approved the final draft.

The following information was supplied regarding data availability:

The raw data is available in the Supplementary Files.

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
