# Peer review of "Unveiling phenolic content, antibacterial, and antibiofilm potential of sacha inchi (Plukenetia volubilis L.) seed shell extracts against Staphylococcus aureus"

_PeerJ, doi:10.7717/peerj.19524_

## Round 0.1 · original submission · Major Revisions

Authors should see all the reviewers' comments; the language needs refinement to remove redundancy and improve flow.

Reviewer 1 ·

Basic reporting

o The language needs refinement to remove redundancy and improve flow.
o Terminology inconsistencies, such as "log reductions," should be addressed.
o Experimental details and references need to be updated and made more precise.
.

Experimental design

o Methodological details are incomplete; for example, the number of replicates and use of controls/blanks are not consistently mentioned.
o Ambiguities such as "1000MIC" should be clarified.
o Comparisons with prior studies or reference standards are necessary to validate findings.

Validity of the findings

o Statistical analysis needs enhancement with post-hoc comparisons and clear significance levels.
o Discussion of the molecular mechanisms underlying biofilm inhibition and bacterial membrane disruption should be expanded.
o The authors should consider suggesting future in vivo studies to validate the observed effects.

Annotated reviews are not available for download in order to protect the identity of reviewers who chose to remain anonymous.

Reviewer 2 ·

Basic reporting

I commend the authors for their extensive data set, compiled over many years of detailed fieldwork. In addition, the manuscript is clearly written in professional, unambiguous language. the article was structured well
I have some comments over all manuscript
If there is a weakness, it is in the statistical analysis (for all tables) which should be improved upon before Acceptance

Experimental design

Methods described well with sufficient details

Validity of the findings

Results and Conclusions are well stated

Additional comments

under the title
Introduction
Several studies have addressed in the literature (in Line 44) the authors must cite several references not one reference Pancu et al., 2021
Method
The reduction capability of DPPH radical is measured by the antioxidant-induced decrease in its absorbance at 515 nm in line168 but in line 175, absorbance at 517 nm was recorded. Please, verify at which absorbance you measured?
Results
Under the title “Chemical composition and antioxidant potential of SI seed shell extract” there is only one reference applied in this section. Please, add many references to support your research for other authors either in the same line or in opposite side
In line 277 the authors said “This might be because the ethanol is more polar “however ethanol is not more polar than water because water is the most polar solvent. But it may be as the authors mentioned later. It might be better solubility for these phytoconstituents
The antioxidant activity evaluated by one method (DPPH), It is preferable to be evaluated by more than one method. Please explain why did you use this method only?
IC50 should written IC50
In table 2 HPLC determination of phenolic compounds in ethanol and aqueous extract of SI seed shell. What is the unit utilized in determination of amount of polyphenolic (μg/ml or what)?
Reference
These references are Not found in manuscript
1. Alibi S, Crespo D, Navas J. 2021. Plant-derivatives small molecules with antibacterial activity.
2. Benfield AH, Henriques ST. 2020. Mode-of-action of antimicrobial peptides: membrane disruption vs. intracellular mechanisms. Frontiers in Medical Technology 2:610997
3. Sathe SK, Kshirsagar HH, Sharma GM. 2012. Solubilization, fractionation, and electrophoretic characterization of Inca peanut (Plukenetia volubilis L.) proteins. Plant Foods for Human Nutrition 67:247-255



If there is a weakness, it is in the statistical analysis (for all tables) which should be improved upon before Acceptance

---

## Round 0.2 · Major Revisions

1. There are several spelling mistakes throughout the manuscript; for example, in abstract, "polyunsaturated" spelling is wrong; "phytocompotents" is found in line no 101;
2. – 50 ”C should be corrected and uniform throughout the manuscript; please check them.
3. "flavanoid" or "flavonoid" line no 149. Correct the sentence line no 177
4. anthocyanins spelling in line 316?
5. anthocyanis ? in line no 322
6. Line no 454 denaturate?
7. Antibiofilm spelling in line no 458 is not correct
Authors should check for any other mistakes throughout the manuscript

In the author response, the authors claim that Language has been improved throughout the text; however, the English needs to be checked thoroughly with the help of an expert, spelling mistakes have to be avoided, and the units used in the manuscript should be uniform throughout the manuscript.

**Language Note:** The Academic Editor has identified that the English language must be improved. PeerJ can provide language editing services - please contact us at [email protected] for pricing (be sure to provide your manuscript number and title). Alternatively, you should make your own arrangements to improve the language quality and provide details in your response letter. – PeerJ Staff

Reviewer 2 ·

Basic reporting

the manuscript is clearly written in professional, unambiguous language. the article was structured well

Experimental design

Methods described well with sufficient details

Validity of the findings

Results and Conclusions are well stated

Additional comments

The authors improved the language and added all the revised comments in the manuscript

---

## Round 0.3 · accepted · Accept

The authors have addressed all of the reviewers' comments, this manuscript is ready for publication.